# COVID-19-Related Effects on the Psychological Well-Being and Resilience of Frontline Expatriate Nurses in the Kingdom of Saudi Arabia: A Qualitative Analysis

**DOI:** 10.3390/healthcare13172200

**Published:** 2025-09-02

**Authors:** Homood A. Alharbi

**Affiliations:** College of Nursing, King Saud University, Riyadh 12372, Saudi Arabia; homalharbi@ksu.edu.sa

**Keywords:** COVID-19, expatriate nurse, frontline nurse, psychological health, pandemic, resilience, Saudi Arabia

## Abstract

**Background:** During the global outbreak of coronavirus disease 2019 (COVID-19), the Kingdom of Saudi Arabia (KSA) ranked second in terms of the highest rate of confirmed cases among the Arab Gulf countries. This situation significantly impacted its population, economy, and healthcare system, especially the psychological well-being of frontline healthcare workers, including frontline nurses. However, there is lack of studies on this topic, particularly on the experiences of frontline expatriate nurses in the KSA, necessitating the utilization of descriptive phenomenological design. **Objective:** This study aimed to explore the experiences of frontline expatriate nurses and their influence on psychological well-being while delivering care to patients with COVID-19. **Methods:** The study employed a qualitative, descriptive phenomenological design that adhered to the key features of Husserl’s phenomenological approach and purposive sampling to select 17 frontline expatriate nurses. They participated in face-to-face semi-structured interviews. Prior to the 17 semi-structured interviews, written consent was obtained from each of the frontline expatriate nurses. The qualitative data collected were analyzed through thematic analysis and rigor was ensured in this qualitative investigation. **Results:** Two primary themes were identified from the interviews: (1) psychological effects due to fear of contracting the corona virus, which included two sub-themes pertaining to organizational stressors resulting in having anxiety, depression, fear, mental stress, and stigmatization; and (2) resilience to cope with the challenges brought by the pandemic, which was characterized by three sub-themes focusing on the importance of accurate information, presence of family and social support, and maintaining good health. The expatriate status of frontline nurses is a key factor influencing in their psychological well-being and resilience. **Conclusions:** Findings indicated that the psychological well-being of frontline expatriate nurses was adversely affected during the pandemic while caring for COVID-19 patients. Consequently, targeted interventions aimed at alleviating the negative psychological impacts of the pandemic are urgently needed to bolster the resilience of frontline expatriate nurses in emergency situations such as the COVID-19 crisis to cope with the associated psychological challenges during the peak of the pandemic, particularly among expatriate nurses.

## 1. Introduction

The COVID-19 pandemic was officially recognized as a global public health emergency in January 2020 [1]. By July 2024, over 775 million confirmed cases, along with more than seven million fatalities, were recorded despite the administration of 5.47 billion vaccine doses worldwide [2]. As reported by the World Health Organization [2], the Kingdom of Saudi Arabia (KSA) has documented approximately 841 thousand cases since the onset of the pandemic, which positions it as the second highest among the six Arab Gulf nations in terms of total recorded COVID-19 cases. Throughout the COVID-19 pandemic, healthcare professionals, particularly nurses, have been identified as a high-risk group for infection [3,4]. According to data collected from 22 July to 15 August 2020, across 37 countries, the WHO reported that 570,000 healthcare workers were infected, with 2500 fatalities attributed to COVID-19 [5]. The WHO classified COVID-19 as a pandemic, which placed healthcare professionals globally in an extraordinary predicament [1]. These workers are on the front lines of the pandemic response and encounter a significant risk of infection. Transmission within hospitals is a critical pathway for the spread of this coronavirus across the globe. Therefore, the psychological well-being and emotional resilience of healthcare workers are vital for sustaining essential healthcare services during the COVID-19 pandemic [6,7]. It is also paramount to ensure enhancement of psychological well-being and emotional resilience in healthcare workers for post-COVID-19 pandemic recovery.

Evidence from a systematic review indicated that Europe experienced the highest total morbidity rates, with 119,628 healthcare workers affected; by contrast, the Eastern Mediterranean region reports the highest mortality rate, at 5.7 per 100 infections [8]. Given the significant impact of the pandemic, measures aimed at safeguarding healthcare workers globally need to be urgently implemented; these measures involve ensuring adequate provision of personal protective equipment (PPE), enhancing preparedness and response strategies, offering training, and ensuring access to food and rest; other measures include addressing issues of fatigue, as well as mitigating psychosocial effects through family support and psychological assistance [4,9,10,11]. Nurses represent the largest segment of healthcare workers within any healthcare institution [12]; they were tasked with delivering patient care, which places them at significant risk of the corona virus infection [3,4]. Research conducted in the KSA by Al Muharraq [13] indicated that out of 215 nurses surveyed, 88% expressed a moral and ethical commitment to care for COVID-19 patients despite the associated risks to their own health.

At the onset, the COVID-19 pandemic led to widespread confinement and movement restrictions, the cessation of economic and educational activities, overwhelmed healthcare systems, and a spectrum of adverse emotional responses (e.g., fear, anxiety, concern, anger, and sadness), all of which have detrimental effects on the overall well-being of the population [14,15]. If these negative emotional states remain unrecognized and untreated, then they may exacerbate psychological distress and adversely impact psychological well-being; these situations further complicate individual health outcomes, particularly among healthcare workers, including nurses in China [16,17], the Philippines [18], Portugal [19], Singapore [20], South Korea [21], and the United States of America (USA) [22]. In the KSA, a study revealed that expatriate nurses, as the predominant participant group, reported experiencing moderate to severe levels of anxiety, depression, and stress during the COVID-19 crisis [23]. The heightened fear of infection contributed significantly to their psychological distress [23]. A notable increase in anxiety, stress, fatigue, and burnout was observed, which was not only due to COVID-19 but also resulted from concerns regarding potential income loss, given the hospital’s “no work, no pay” policy [23]. In addition, many expatriate nurses experienced the absence of family in the KSA due to policy restrictions as a missing support system, adding to their loneliness and psychological stress during the pandemic [23].

There are significant insights that pertain to specific possible improvements in the Saudi Arabian context, particularly in preparation for future pandemics, especially among expatriate nurses, though not exclusively limited to the following examples. The first important insight pertains to amendments in labor policies for expatriate healthcare workers, such as permitting family accompaniment or visits during crises to provide emotional support and improve morale. The second significance of conducting this study is pertinent to mental well-being and coping strategy modules that may be integrated into continuing professional education for nurses and paramedics, aimed at raising awareness and equipping them with self-care techniques. Furthermore, public health awareness and support should be directed towards fostering respect and support for frontline healthcare workers, especially expatriate nurses, to mitigate social stigmatization, a prevalent mental health issue during the pandemic among healthcare workers, as reported in various countries, including Egypt [24], Iran [25], Italy [26], Jordan [27], Korea [28], Taiwan [29], the USA [30,31], and the KSA [32,33,34,35]. Considering these factors from international and national perspectives, it is essential to delve deeply into the experiences of expatriate nurses that impact their psychological well-being amid the COVID-19 pandemic. Consequently, this study aims to examine the experiences of frontline expatriate nurses and their effects on psychological well-being while delivering care to COVID-19 patients in a university hospital located in Riyadh, KSA. Based on this, it was appropriate to employ and adhere to the key features of Husserl’s phenomenological approach [36], which significantly contributed to achieving the aim of the current study.

### Aim of the Study

This study aimed to explore experiences related to the psychological well-being of frontline expatriate nurses delivering care to COVID-19 patients at a university hospital in Riyadh, KSA. It specifically sought to address the following research questions: (1) How do frontline expatriate nurses experience and make sense of the psychological impact of caring for COVID-19 patients in the KSA? (2) How do frontline expatriate nurses navigate psychological challenges and foster resilience while caring for COVID-19 patients? (3) How does the expatriate status of frontline expatriate nurses shape their psychological experiences and access to resilience-building resources during the COVID-19 pandemic in the KSA?

## 2. Materials and Methods

### 2.1. Research Design

This study utilized a qualitative, descriptive phenomenological design of research, which followed the essential features of the Husserlian phenomenological approach [36]. The design used words, narrative, and in-depth conversations from the participants as the main source of data. This process involved knowing, exploring, and formulating meaning of the participant’s responses through one-on-one interview or in-depth conversations with regard to psychological well-being while caring for COVID-19-infected patients. In this study, the inclusion and utilization of any theoretical framework were not considered in alignment with Husserl’s stance that no a priori phenomenological or theoretical framework should direct the phenomenological investigation [36,37].

### 2.2. Sample and Sampling Technique

This research utilized a non-probability purposive sampling method, whereby participants were selected according to specific inclusion criteria established in the current study. In this approach, the following criteria were defined and served as a guide for selecting participants for interviews. Participants were classified as frontline expatriate nurses if they met the following conditions: (a) they were registered nurses; (b) they were assigned to the COVID-19 ward, emergency department, fever clinic, or flu clinic; (c) they had been involved with or exposed to COVID-19 patients for a minimum of 3 months while working at least 40 h per week; and (d) they were non-Saudi or expatriate nurses. These individuals were selected for the study due to their significant role in examining the psychological well-being of frontline expatriate nurses providing care to COVID-19 patients during the pandemic. For qualitative research, Creswell [38] recommended a sample size of 5 to 25 participants, while Holloway and Galvin [39] suggested a range of 8 to 12 participants as suitable. Consequently, the researcher recruited 17 frontline expatriate nurses for this study, which allowed for data saturation, defined as the point at which no new themes emerged [40,41], after interviewing the 17th participant. In this study, data saturation was reached when no new codes emerged, and subsequent interviews conducted after the 17th participant produced redundant information regarding key themes. The sample was intentionally chosen to include frontline expatriate nurses who were actively involved in patient care during this period.

### 2.3. Ethical Considerations

Prior to initiating the 17 semi-structured interviews, ethical approval was secured from the Institutional Review Board of the university hospital located in Riyadh, KSA. The participants were provided with informed written consent to ensure they comprehended the objectives of the study and their volunteered participation. Given the sensitive nature of the study, which examined the psychological well-being of frontline expatriate nurses exposed to COVID-19 patients, the researcher assured participants that their identities and the names of their respective hospitals would remain confidential. Participants were also informed that the interviews would be recorded. The recordings would be stored securely in a drive accessible solely to the researcher for transcription purposes and subsequently be deleted after transcription. Each participant was interviewed solely regarding their experiences related to psychological well-being and its implications during the COVID-19 pandemic.

### 2.4. Data Collection Process

Following the acquisition of ethical approval, the researcher ensured adherence to the established inclusion criteria to assess the eligibility of the frontline expatriate nurses. Each participant who met the eligibility criteria was approached for written consent. This form provided a detailed explanation of the nature, ethical considerations, potential benefits, and associated risks of the study. Specifically, an email invitation along with a participant information sheet and informed consent form were disseminated to the frontline expatriate nurses by their department manager through their hospital email addresses, subsequent to the application of the inclusion criteria. The invitation included the contact information of the researcher (i.e., email address and mobile number). It encouraged interested participants to reach out for involvement in the interview.

In this study, the semi-structured interviews with the 17 frontline expatriate nurses were conducted face to face while adhering to COVID-19 precautionary measures and social distancing. Once participants consented to the terms, the researcher initiated the recorded interview process, which considered the availability of the participants. The researcher was responsible for scheduling the interviews and maintained exclusive access to all participant responses. Data collection involved semi-structured interviews, which enabled participants to discuss their experiences and circumstances related to providing nursing care for COVID-19 patients, particularly concerning their psychological well-being. Interruptions from the researcher were also minimized during each interview.

The following is the topic guide utilized for the 17 semi-structured interviews.

(i)How do frontline expatriate nurses experience and make sense of the psychological impact of caring for COVID-19 patients in the KSA?
-Could you share your experiences related to working on the frontline and delivering care to patients suspected or confirmed to have contracted the coronavirus during the COVID-19 pandemic?-How do you describe your nursing care of patients suspected of or diagnosed with COVID-19 infection?-What are your feelings when providing care for a patient suspected of or diagnosed with COVID-19?-What are your concerns, fears, and anxieties related to caring for or being exposed to COVID-19 patients?-How did the COVID-19 pandemic cause stress in your life and did it also dramatically affect you and your family abroad in your home country?-What are the issues and challenges you encountered in caring for patients suspected of or diagnosed with COVID-19 infection?(ii)How do frontline expatriate nurses navigate psychological challenges and foster resilience while caring for COVID-19 patients?
-How has the COVID-19 pandemic influenced your temperament, emotions, and feelings regarding the management of your psychological challenges?-What actions did your hospital undertake to assess, identify, or address the effects of the COVID-19 pandemic on your psychological well-being?-What should healthcare workers or administrators do for the psychological well-being challenges experienced related to COVID-19?-How did your hospital organization consider the psychological effects of the COVID-19 pandemic on the mental health, well-being, and quality of life of expatriate nurses?(iii)How does the expatriate status of frontline expatriate nurses shape their psychological experiences and access to resilience-building resources during the COVID-19 pandemic in the KSA?
-How do you cope with your psychological well-being issues while caring or being exposed to COVID-19-infected patients?-What mental health services, resources, and support did you have throughout the COVID-19 pandemic?-What ways or strategies did you undertake to maintain your own protection against COVID-19 and to maintain your resilience and psychological well-being?-What are your thoughts on the methods to be employed by hospital organizations to support and protect the psychological well-being of expatriate nurses in preparation for future pandemics?

### 2.5. Rigor

This study meticulously ensured rigor in this qualitative investigation, which guaranteed that the gathered data were robust and reliable for the audience. The initial crucial step undertaken by the researcher pertained to personal reflexivity that involved reflecting on the typical routines when working in the intensive care unit (as a previous job before joining academia) and addressing assumptions around how frontline expatriate nurses, who could be former colleagues, think about and experience caring for patients suspected or confirmed to have COVID-19 infection. During data generation and analysis, the researcher started to uncover various nuances associated with the psychological effects of COVID-19 during the peak of the pandemic when providing care to patients with COVID-19, whether suspected or diagnosed.

According to Maher et al. [42], trustworthiness serves as a more suitable standard for assessing qualitative research. To establish a trustworthy process, this study will implement the “four-dimensional criteria” proposed by Guba and Lincoln [43]. These criteria encompass credibility, dependability, confirmability, and transferability. The initial criterion, that is, credibility, was addressed through a structured interview process with participants. The researcher executed five pilot interviews to enhance the overall methodology, which focused on the interview protocol, time management, and the seamless execution of interviews. In addition, credibility was supported by member checking by the assistant researcher. This process ensured the accuracy of data collection and consistency among participants.

Subsequently, the criterion of dependability was introduced to guarantee consistent data collection devoid of unnecessary fluctuations. The researcher employed an audit check strategy to meticulously document the methodologies of data gathering, analysis, and interpretation; this approach rendered the process repeatable and auditable for future researchers to comprehend the decision-making trail, the procedural steps, and the resultant findings [44]. An audit trail was then established to maintain a comprehensive record of the data collection process, which guaranteed confirmability. During the interviews, recordings were made, and the researcher systematically compared the transcripts with the audio files to verify the accuracy and clarify the data obtained from participants. In the end, the criterion of transferability was addressed in this research by compiling the responses and observations of the participants involved. The researcher aimed to produce a rich (thick) description of the findings of the study, which were presented through a supported by detailed contextual descriptions and a demographic table (see Table 1). In this way, readers could evaluate the applicability of the results to their own contexts.

### 2.6. Data Analysis

In this study, thematic analysis was employed as a methodological approach to analyze the collected qualitative data. This study adheres to the six phases of thematic analysis: (1) becoming acquainted with the data, (2) generating initial codes, (3) identifying themes, (4) reviewing themes, (5) defining and naming themes, and (6) compiling the report [45,46]. NVivo software Version 14 was utilized to manage the qualitative data, and the analysis was conducted exclusively in the English language. Thematic analysis is fundamental to qualitative methodology, and the themes were validated through the use of word clouds generated by NVivo software [47]. A subset of transcripts was coded independently by two researchers (the researcher and an assistant researcher). Discrepancies were discussed until consensus was achieved.

## 3. Results

A total purposive sample of 17 frontline expatriate nurses participated in the semi-structured interviews. The participants’ average age was 37.3 years, with majority of them being female, married, and Indians, with highest proportion working in the intensive care unit (see Table 1).

Overall, the participants reported that managing significant responsibilities while working on the frontlines was mentally taxing during the COVID-19 pandemic. They encountered numerous psychological well-being challenges originating from media influences, as well as obstacles related to their work environment during the pandemic. In addition, shifts in teamwork dynamics and collaboration while caring for COVID-19 patients complicated their experiences. The participants addressed the psychological impacts of their hospital work during this period by utilizing various coping strategies. Specifically, the qualitative findings of this study revealed that frontline expatriate nurses actively and collaboratively engaged in semi-structured interviews, which led to the identification of two primary themes derived from the collected data, as summarized below.

The two primary themes include (1) psychological effects due to fear of contracting the corona virus and (2) resilience to cope with the pandemic. The primary themes were further substantiated by sub-themes and direct quotations from the interviews with the participants. For the first primary theme, psychological effects experienced by the participants while working in the hospital during this period, two sub-themes emerged, including (1) experiencing organizational stressors such as anxiety, depression, fear, and mental stress, and (2) stigmatization. For the second primary theme, the participants highlighted resilience through the utilization of their coping mechanisms, having three sub-themes that emerged, including (1) obtaining accurate information, (2) receiving family and social support, and (3) maintaining good health.

### 3.1. Major Theme 1—Psychological Effects Due to Fear of Contracting the Corona Virus

The first primary theme identified in this research pertains to the psychological impacts experienced by participants as a result of the COVID-19 pandemic while caring for or being in contact with patients infected with the virus. These psychological impacts encompass the sub-themes, including experiencing organizational stressors (feelings of anxiety, depression, fear, mental stress) and stigmatization associated with providing care to COVID-19 patients. When participants recounted their experiences of caring for or being exposed to these patients, they expressed similar sentiments. Frontline expatriate nurses indicated that they maintained the same level of nursing care for all patients during the pandemic, regardless of whether they were infected or merely suspected of having the virus. However, they could not escape from these organizational stressors of having feelings of fear, anxiety, or nervousness regarding the possibility of contracting the virus and subsequently facing isolation during recovery. Frontline expatriate nurses residing with their families in the KSA had an additional concern of potentially transmitting the virus to their family members upon returning home from work. Meanwhile, those who did not have their families with them in the KSA experienced distinct challenges of not being able to communicate with their families in their home country. The experiences shared by the participants highlight the significant effects of the pandemic on their psychological well-being.

Participants expressed feelings of anxiety in their professional environment. For these nurses, the workplace served as a second home. They articulated a strong sense of duty in their role of providing nursing care to any patient seeking consultation or hospitalization. During the pandemic, participants reported heightened anxiety, particularly Nurse Expat 5, who remarked, “Anxiety is present, but we must manage it because patients require our care, and we must deliver what is best for them.” Similarly, Nurse Expat 12 conveyed, “When I am in contact with a patient diagnosed with COVID-19, I experience nervousness and anxiety due to the risk of potential infection.” Furthermore, Nurse Expat 15 noted, “I felt apprehensive about contracting COVID-19 and the prospect of being alone or isolated if I were to become infected.”

During the COVID-19 pandemic, nurses reported experiencing depression, which was a sentiment echoed by participants in this study who also indicated feelings of depression while caring for COVID-19 patients. They articulated the impact of their nursing responsibilities on their mental well-being, which revealed fears of contracting the virus and the emotional toll of quarantine and isolation. The accounts of the participants reflect the psychological challenges encountered by nurses during this crisis. For instance, Nurse Expat 7 expressed feelings of depression, noting, “Wearing PPE all day is challenging, and I often feel judged and isolated by community members who are concerned for their own safety if I were to contract the virus.” Similarly, Nurse Expat 1 highlighted, “Frontline healthcare workers are susceptible to short- and long-term psychological effects, including depression, which came from the anxiety of potential infection and the implications of quarantine, as well as concerns for the health of their family and friends.”

Evaluating the anxiety experienced by nurses during the COVID-19 pandemic is essential for understanding its psychological well-being implications and effects. This knowledge is crucial as the psychological strain of fear may lead to suboptimal clinical decision making and adverse patient care outcomes, which ultimately impact the job performance and professional development of nurses. Participants in this study consistently expressed feelings of fear regarding potential exposure to and infection with COVID-19. They also reported concerns about being isolated from their families, particularly for those with relatives in the KSA. During the interviews, Nurse Expat 4 articulated a fear of transmitting the virus to family members, friends, or colleagues, stating, “[I have] the fear of being infected with COVID-19 and also, being a carrier of the disease and transmitting it to my family, friends, and colleagues.” Similarly, Nurse Expat 6 expressed, “In general, [I have] fear of failing to meet our family, friends, and relatives [during the pandemic].” Nurse Expat 17 remarked, “While fear is present, I believe caring for patients with COVID-19 is a valuable experience because it can enhance my professional growth. I also fear being separated from my family and the possibility of infecting others (colleagues, friends).” Furthermore, Nurse Expat 11 conveyed a fear of hospitalization if infected, stating, “I fear for the possibility of being infected, and if that happens, then experiencing severe symptoms or requiring hospitalization.”

The study participants reported experiencing mental stress as a significant psychological effect. They expressed various perspectives and experiences related to their stress, which originated from concerns regarding the risk of infection, the potential for isolation or hospitalization, and the emotional toll of being separated from family due to canceled vacations. The participants articulated the factors contributing to their stress. For instance, Nurse Expat 1 noted feeling mentally stressed, stating, “We encountered psychological stress due to increased workloads, inadequate quality PPE, social exclusion and stigmatization, lack of incentives, and poor coordination and management during the pandemic.” Similarly, Nurse Expat 13 remarked, “I believe caring for patients with COVID-19 is a source of stress.” Nurse Expat 3 added, “Nurses commonly experience stress, especially when they are aware that patients are COVID-19 positive, compounded by the effects of isolation or lockdown.” Nurse Expat 9 shared about not having family members in the KSA, “I was unable to reach out to my family especially my kids as I was working 12-h shifts on certain days during the peak of the pandemic. Consequently, it was quite challenging to consider the lack of time to communicate with my family back in my home country.”

The social stigma associated with the COVID-19 pandemic represents a significant factor influencing the mental well-being of nurses, given that it contributed to an increased risk of virus transmission during the peak of the pandemic. Participants in this research expressed experiences of social stigma from their friends and the local community. For instance, Nurse Expat 8 remarked, “Frontline nurses like me, are susceptible to psychological challenges, including feelings of stigmatization and rejection from those in our neighborhoods.” Similarly, Nurse Expat 10 noted, “Regardless of whether my patients are infected, I provide the same level of care. However, I encounter stigma when I am in the community.”

### 3.2. Major Theme 2—Resilience to Cope with the Challenges Associated with the Pandemic

This research highlighted the second primary theme, which is related to the resilience demonstrated by frontline expatriate nurses throughout the pandemic. The resilience of nurses was evident through various coping strategies, identified as sub-themes such as (1) acquiring reliable information regarding the pandemic, (2) garnering support from family and social networks, and (3) emphasizing personal health.

For the first sub-theme, the participants expressed the necessity of gathering correct information about the pandemic to manage psychological distress while ensuring their safety by adhering to health protocols against COVID-19. These protocols encompass social distancing, practicing hand hygiene, wearing masks, and crucially, getting vaccinated. This sub-theme is supported by the experiences of expatriate nurses, including Nurse Expat 11, who emphasized, “Healthcare workers and administrators need to attend lectures or seminars on COVID-19 for enhancing their knowledge and utilize PPE properly.” In addition, Nurse Expat 14 noted, “Nurse educators must offer correct and valuable information on caregiving and preventive measures related to COVID-19.”

In the second sub-theme, the participants involved in this research identified familial and social support as essential mechanisms for coping during the pandemic. Nurse Expat 8 notably articulated that “the implementation of social distancing has resulted in fewer opportunities for social engagement and participation in events. Addressing social support through policy initiatives is often crucial. Social support systems play a vital role in protecting healthcare professionals and reducing the incidence of psychological distress.” Furthermore, Nurse Expat 16 highlighted the significance of family during the pandemic, particularly while being away from their home country, stating, “We need to consistently communicate with our family and loved ones through video calls to alleviate our stress and receive comfort.”

The third sub-theme identified by nurses as a coping strategy is the maintenance of health, which encompasses spiritual well-being through regular prayer and other faith-based practices. Participants indicated that health encompasses not only physical, emotional, and mental well-being but also spiritual well-being. In this context, Nurse Expat 2 highlighted, “Adhering to safety protocols against COVID-19 significantly contributes to my safety and health. I also consistently consume vitamins and maintain a nutritious diet to navigate this health crisis.” Additionally, Nurse Expat 2 remarked, “To manage the challenges posed by COVID-19, I require ongoing communication with my family and the practice of prayer.” Nurse Expat 12 noted, “I manage anxiety and fear by receiving two vaccine doses, ensuring my physical and psychological well-being, consuming nutritious foods, and exercising every off day.” Similarly, Nurse Expat 9 remarked, “To address anxiety and fear related to the COVID-19 pandemic, I prioritize connecting with supportive individuals, ensuring adequate sleep, taking vitamins, and eating a balanced diet.” Finally, Nurse Expat 15 stated, “Maintaining health through a proper diet and exercise, minimizing exposure to crowds, and completing the vaccination regimen have been essential in coping with the pandemic.”

In this study, the researcher presents a simple conceptual framework based on the findings that would be a valuable contribution to the existing nursing body of knowledge. This framework visually represents the interplay between pandemic-related (organizational) stressors, the psychological effects on expatriate nurses, the moderating role of their expatriate status, their coping strategies/resilience factors, and the resulting psychological well-being outcomes (see Figure 1).

## 4. Discussion

This study explored the psychological well-being challenges experienced by frontline expatriate nurses providing care to COVID-19 patients at a university hospital located in Riyadh, KSA during the peak of the pandemic. Similar findings were observed in a comprehensive analysis of the literature, which indicated that a scoping review regarding the psychological effects of the COVID-19 pandemic on emergency healthcare workers revealed elevated levels of anxiety, burnout, depression, and stress among nurses, in contrast to other healthcare professionals within hospital emergency departments [48]. Beyond these psychological challenges, the review by Alanazi et al. [48] highlighted that emergency nurses are also susceptible to severe psychological repercussions originating from insufficient sleep, symptoms of post-traumatic stress disorder, as well as psychological distress and secondary trauma associated with their roles during the pandemic. In a national cross-sectional online survey conducted in China by An et al. [49], the prevalence of depression among nurses (n = 1103 emergency nurses) was examined, which revealed an overall depression rate of 43.61%. Another cross-sectional study in China identified that the fear of transmitting the virus to family members was a significant predictor of anxiety and stress among nurses working in emergency departments and fever clinics during the pandemic. Furthermore, a study assessing the mental health of 14,825 emergency department medical staff, including nurses, found that the nursing profession was linked to a heightened psychological impact from the infection, particularly in terms of post-traumatic stress disorder [50]. Moreover, Huerta-González et al. [51], in their review paper, emphasized that frontline care for COVID-19 patients has profound psychological effects on nurses. They obtained this conclusion from their synthesis of evidence from 13 qualitative studies exploring the perceptions of nurses on the psychological impacts of treating hospitalized COVID-19 patients [51].

The literature extensively documents the anxiety experienced by nurses as a result of the COVID-19 pandemic. For instance, two studies conducted in Turkey revealed that nurses among emergency healthcare workers exhibited higher anxiety levels than their counterparts in other emergency roles [52]. Furthermore, these nurses expressed concerns regarding the availability of adequate PPE and reported anxiety related to the support received from management and the timeliness of salary payments [52]. In addition, a Turkish study and a Libyan study found that participants experienced significant anxiety originating from fears of transmitting the virus to family members [53,54]. Similarly, research in China by Cui et al. [55] identified the fear of infecting family members as the primary predictive factor for anxiety. In addition to countries such as China, Libya, Turkey, and the United States, a widespread global concern exists regarding the anxiety experienced by nurses who are in close contact with COVID-19 patients. This phenomenon has been particularly documented in various regions, including Africa [56], Australia [57], Iran [58,59], Portugal [60], the KSA [61,62], Spain [63], and among nursing professionals in the Philippines [18].

In addition, a quantitative investigation conducted in China indicated that the overall prevalence of depression among nurses was 43.61%. Factors such as employment in tertiary hospitals, direct patient care for COVID-19 patients, and current smoking status were found to be significantly correlated with depression [49]. In Turkey, another study highlighted that healthcare professionals, particularly nurses in the emergency department, reported experiencing severe depression, which was identified as a significant risk factor for mental health issues [53]. In the KSA, a descriptive cross-sectional study involving 323 frontline healthcare workers, predominantly non-Saudi (expatriate) nurses, revealed that non-Saudi healthcare workers experienced higher levels of depression than their Saudi counterparts; moreover, nurses reported greater levels of depression than their physician colleagues [64].

The context of fear as a psychological effect of COVID-19 during the peak of the pandemic as reported by the study participants echoed a similar finding among nurses in China, who expressed fears of infection originating from a lack of understanding regarding the behavior of the virus [55,65]. In the KSA, a cross-sectional study involving 969 participants revealed that the fear experienced by nurses, predominantly non-Saudi (i.e., expatriate nurses), may be attributed to various factors, including the rapid increase in COVID-19 cases and fatalities within the country [66]. Another contributing factor is the role of nurses as frontline workers directly engaged in the treatment of COVID-19 patients, which heightens their fears on becoming infected and transmitting the virus to others, including family members [66]. Finally, the fear of the unknown, particularly during the early stages of the outbreak with no specific treatments and limited knowledge regarding the transmission of the disease, has been a major concern [66]. In addition, Huerta-González et al. [51] asserted that the predominant fear among nurses is related to the possibility of infecting family members or contracting the virus themselves. As the pandemic has evolved, this fear has overshadowed the stress that was initially the primary psychological impact experienced during the early phases of the crisis [51].

Numerous investigations have demonstrated that frontline care for patients admitted with COVID-19 infection has resulted in various psychological effects on nurses, with stress being a prominent concern as one of the sub-themes of the psychological effects of COVID-19. This finding is consistent with a study in Turkey where nurses exhibited higher stress levels than other emergency personnel, such as physicians, security staff, those involved in patient transport, cleaning personnel, and data entry staff. Importantly, the stress levels among nurses were significantly elevated in contrast to those of physicians, with perceived stress identified as a critical risk factor for mental health [52,53]. A study conducted in China revealed that the fear of transmitting the virus to family members was a key predictor of stress among nurses working in emergency departments and fever clinics [55]. Another study in China by Li et al. [67] found a correlation between stress symptoms and PTSD symptoms. In a qualitative study by Munawar and Choudhry [68], participants indicated that, during the pandemic, media coverage was perceived as a primary contributor to heightened stress levels, as the reliability of the information disseminated was often questionable. In the KSA, a quantitative study involving 469 registered nurses across various hospitals was conducted from July to September 2020 [69]. The findings revealed that over a quarter and nearly half of the nurses surveyed experienced high and moderate levels of stress, respectively [69].

Stigmatization is also another sub-theme of the psychological effects reported by frontline expatriate nurses in this study. The presence of social stigma among healthcare professionals, including nurses, can significantly impact their mental well-being and professional performance, as evidenced in past pandemics [70] and during the COVID-19 pandemic [71]. Nurses, as vital components of the multidisciplinary healthcare team, encounter challenges on the front lines and may encounter stigma, which complicates efforts to manage epidemic diseases and adversely affects their mental health [72]. According to the WHO [2], the stigma associated with the ongoing COVID-19 pandemic may exacerbate the likelihood of virus transmission. Numerous investigations have documented the prevalence of social stigmatization worldwide. For example, a qualitative study examined the experiences of Italian nurses regarding perceived stigma during the COVID-19 pandemic utilizing the phenomenological Cohen method [72]. The analysis revealed two primary themes: (1) stigma within the workplace and (2) stigma in daily life [72]. Each theme encompassed several sub-themes, including “perceptions akin to gun sights,” “avoidance of physical proximity,” “a reluctance to engage physically,” and “the burden of familial association.” Simeone et al. [72] posited that public health crises, such as the COVID-19 pandemic, represent significant stressors for individuals and communities; thus, stigma may pose a greater threat than the illness itself and serve as a substantial barrier to effective medical and psychological care. The findings of Simeone et al. [72] suggest that comprehending the experience of stigma is essential for developing targeted intervention programs designed to safeguard the mental well-being of healthcare professionals. A separate investigation explored the social stigma linked to the COVID-19 pandemic across various social groups within Indian society, which revealed an increase in discrimination based on race, class, and religion [73]. The findings underscored that individuals identified as infected or as close contacts of infected persons, particularly healthcare workers such as nurses, encountered stigma related to their race, religion, and social class during the pandemic [73]. In the KSA, a qualitative analysis focused on the stigma encountered by nurses caring for COVID-19 patients. Through individual interviews, nurse participants identified four primary themes and three sub-themes [74]. These themes encompassed (1) the labeling of nurses as “COVID nurses,” which was accompanied by a sub-theme of frustration; (2) apprehension regarding the unknown, with a sub-theme of uncertainties; (3) the necessity for support among nurses; and (4) a passion for their profession, which was highlighted by a sub-theme of nurses’ self-worth. The research conducted by Pasay-An et al. [74] concluded that nurses providing care for COVID-19 patients encountered significant stigma, and these nurses were predominantly being referred to as “COVID nurses.”

The findings concerning the resilience of frontline expatriate nurses to cope with the challenges associated with the pandemic are closely linked to the coping strategies employed by participants in this study. This result is consistent with a narrative literature review that examined eleven studies (two qualitative and nine quantitative) focused on the coping mechanisms adopted by frontline nurses during the COVID-19 pandemic [75]. This review identified six significant themes: (1) the implementation of protective measures against COVID-19, (2) avoidance strategies, (3) social support, (4) faith-based practices, (5) psychological support, and (6) management assistance [75]). Sehularo et al. [75] suggested that these six coping strategies may help mitigate the stress and burnout experienced by nurses during the pandemic. Similarly, social support, along with three additional critical themes—self-support in times of crisis, familial protection from a distance, and the impact of the hospital environment—were identified as coping strategies by 16 Iranian nurses in a qualitative study utilizing a descriptive phenomenological methodology [76]. Furthermore, a qualitative exploratory investigation examined the mental health and coping strategies of nurses working with suspected and confirmed COVID-19 patients in Brunei Darussalam [77]. The study involved 11 focus group discussions conducted from October 2020 to January 2021, with participation from 75 nurses. Three key themes emerged: (1) “COVID-19 roller-coaster transitional journey,” which illustrated the varied psychological responses of nurses as they transitioned from merely hearing regarding COVID-19 to experiencing its impact firsthand during the community outbreak; (2) “mind my mind and heart,” which conveyed the experiences of nurses regarding their mental health and emotional reactions; and (3) “the psychosocial system,” which outlined the coping strategies utilized by nurses throughout the COVID-19 pandemic [77]. The findings of Maideen et al. [77] indicated that nurses adopted psychosocial coping strategies at various stages of the pandemic, which relied on support from family, friends, the public, and governmental resources to navigate the challenges posed by the crisis. In addition, the availability of vaccines against the COVID-19 infection greatly helped the participants to cope with the psychological challenges posed by the pandemic, particularly during its peak and its long-term psychological impact post-pandemic period. This finding is consistent with previous studies in Indonesia [78], the United Kingdom [79], and in Pandey’s review paper [80].

As in all qualitative research, findings of the current study are context-bound and may not be generalizable. The researcher’s positionality and reflexivity are acknowledged as influencing data collection and interpretation.

## 5. Implications and Recommendations

In light of the necessary modifications and enhancements in nursing practices related to the psychological well-being and resilience of frontline expatriate nurses caring for patients suspected or confirmed to have COVID-19, as well as in anticipation of future pandemics, several recommendations are presented herein. This study emphasizes the importance of maintaining resilience in all circumstances because frontline expatriate nurses are inherently resilient. Nurses, as the primary caregivers during a pandemic, particularly COVID-19, can employ various positive coping strategies to navigate mental challenges. These strategies would help safeguard their mental well-being while remaining resourceful, adaptable, and flexible. In addition, nurses should consistently engage in and value the educational training and psychological support provided by their hospitals, which are crucial for fostering a highly supportive work environment. Targeted interventions aimed at enhancing the coping strategies of frontline expatriate nurses should also be implemented. These interventions should include orientation sessions and workshops that address COVID-19 infections, continuous updates on COVID-19 restrictions and protocols, and organizational support and guidance. Such support must encompass manageable (flexible) nursing workloads, suitable staffing schedules, sufficient medical equipment and supplies, and resources that cater to the psychological well-being requirements of frontline expatriate nurses, such as better communication channels and mental health days.

Nursing administrators can utilize the insights gained from this research by recognizing the various threats to the psychological well-being of nurses originating from the COVID-19 pandemic, especially those associated with the work environment and infection-related concerns. By implementing these findings, they can enhance the availability of PPE, establish effective testing protocols, ensure adequate staffing levels, promote equitable scheduling and rotation of nursing staff, and foster clear communication regarding COVID-19 guidelines. These initiatives are crucial for mitigating the psychological impact experienced by nurses throughout the pandemic. Moreover, nursing administrators can employ the suggested intervention plan to address psychological well-being challenges, overcome obstacles, and combat the social stigma encountered by the participants.

In light of the findings, hospital administrators must acknowledge the significant psychological effects on frontline expatriate nurses by instituting regular mental health assessments and broadening these services to encompass other healthcare personnel impacted within the institution. This strategy can aid hospital administrators in developing preventive measures designed to mitigate the risk of mental health challenges. Such measures may encompass prompt financial support, access to psychological counseling, on-site mental health assistance, and the provision of psychiatric care for nurses who are especially susceptible due to their direct engagement in the treatment of COVID-19 patients. Furthermore, future pandemic response strategies and protocols must be designed to bolster frontline expatriate nurses and other healthcare workers. These strategies would alleviate the psychological repercussions of the pandemic. These strategies and protocols should prioritize the mental health of the nursing workforce through emphasizing the necessity for referrals to specialized mental health services, including the availability of psychologists, counselors, or psychiatrists.

The implications of the findings of this study could be considered by refining educational goals that integrate theoretical knowledge and practical clinical training. This strategy should focus on equipping nursing students and practicing nurses with essential knowledge and skills to safeguard against infections and support their mental health during challenging times, such as the COVID-19 pandemic. Furthermore, nurse educators ought to include discussions on the role of social stigma and the significance of resilience. The application of coping mechanisms in crisis situations such as the COVID-19 pandemic should be emphasized while also highlighting critical information pertinent to the topic. Future research must recognize that the findings of this study highlight the urgent need for further exploration into the psychological well-being challenges and resilience of frontline expatriate nurses. Utilizing quantitative surveys could help validate and strengthen the qualitative findings of this study. A larger sample size should be employed to enhance the applicability of the results to frontline expatriate nurses in diverse healthcare environments, including public and private sectors, as well as various regions in the KSA and other Arab Gulf countries. Subsequent research should focus on identifying and analyzing confounding variables that may affect the psychological well-being and resilience utilizing available coping strategies of frontline expatriate nurses during crisis situations such as the COVID-19 pandemic. Importantly, broader policy implications are vital to this vulnerable group of frontline expatriate nurses such as the need for national or institutional policies that specifically address the well-being and support of expatriate healthcare workers, particularly nurses during public health emergencies as well as possible amendments in labor policies, including permit for family accompaniment or visits during health crises which could provide emotional support and improve the morale of expatriate healthcare workers.

## 6. Conclusions

A thorough exploration of the experiences of frontline expatriate nurses in delivering care to COVID-19 patients is crucial because their responses reveal significant psychological well-being challenges encountered during the pandemic. The findings of this study have shed light on the psychological well-being of these nurses, which demonstrated that they encountered various psychological impacts due to organizational stressors, including anxiety, depression, fear, and mental stress, and societal stigmatic beliefs while caring for patients infected with COVID-19. Furthermore, the study emphasizes the resilience to cope with the associated challenges brought by COVID-19, as exhibited by frontline expatriate nurses in addressing the psychological ramifications of the pandemic, which was achieved through access to accurate information, support from family and social networks, and maintenance of their overall health. Frontline expatriate nurses were highly vulnerable to the psychological effects during the peak of the pandemic, necessitating that targeted interventions for this group related to their psychological well-being are particularly crucial to cope with and navigate the challenges associated with the COVID-19 pandemic and in preparation for future pandemics.

## Figures and Tables

**Figure 1 healthcare-13-02200-f001:**
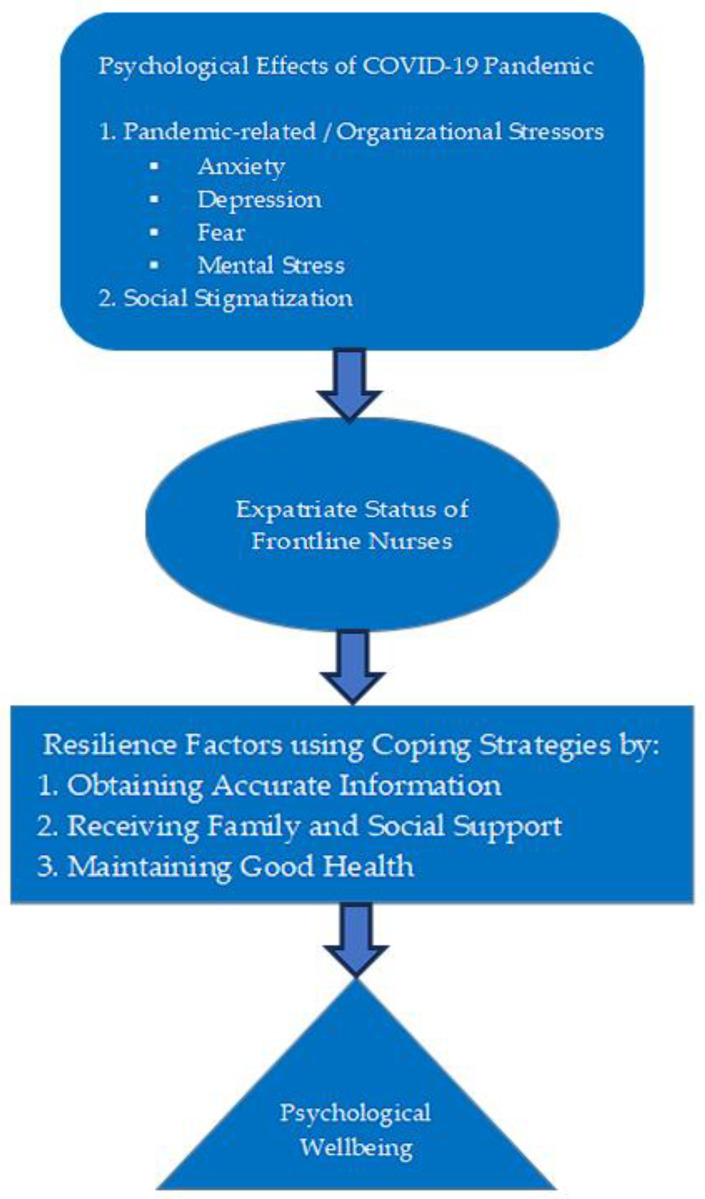
Conceptual framework based on the study findings.

**Table 1 healthcare-13-02200-t001:** Demographic profile of participants.

Participant Code	Age (In Years)	Gender	Civil Status	Nationality	Assigned Area
Nurse Expat 1	33	Male	Single	Filipino	COVID ward
Nurse Expat 2	41	Female	Married	Jordanian	Intensive care unit
Nurse Expat 3	28	Male	Single	Filipino	Flu clinic
Nurse Expat 4	35	Female	Married	Filipino	Emergency department
Nurse Expat 5	33	Female	Married	Indian	Flu clinic
Nurse Expat 6	52	Female	Married	Indian	Intensive care unit
Nurse Expat 7	38	Female	Married	Filipino	Intensive care unit
Nurse Expat 8	39	Female	Married	Indian	Emergency department
Nurse Expat 9	34	Male	Married	Indian	Intensive care unit
Nurse Expat 10	38	Female	Married	Jordanian	COVID ward
Nurse Expat 11	43	Female	Married	Indian	COVID ward
Nurse Expat 12	40	Male	Single	Indian	Emergency department
Nurse Expat 13	35	Female	Married	Filipino	Intensive care unit
Nurse Expat 14	36	Male	Married	Indian	Intensive care unit
Nurse Expat 15	28	Female	Single	Indian	Emergency department
Nurse Expat 16	44	Female	Married	Filipino	Intensive care unit
Nurse Expat 17	37	Female	Married	Jordanian	Flu clinic

## Data Availability

The data presented in this study are available on request from the corresponding author. This study adhered to the COREQ (Consolidated Criteria for Reporting Qualitative Research) checklist for qualitative studies [81].

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
