# Peer review of "COVID-19-Related Effects on the Psychological Well-Being and Resilience of Frontline Expatriate Nurses in the Kingdom of Saudi Arabia: A Qualitative Analysis"

_healthcare, 2025, doi:10.3390/healthcare13172200_

Round 1

Reviewer 1 Report (Previous Reviewer 1)

Comments and Suggestions for Authors

I thank the author for taking the time to revise their manuscript. They have addressed most of the comments from the first round of revision. However, some minor amendments are still required.

  1. In the introduction, the author now provides more context on the COVID-19 pandemic. However, now that the pandemic has ended, the introduction should reflect why it is important to ensure psychological well-being and emotional resilience in healthcare workers for post-pandemic recovery (e.g., lines 53-55).
  2. When structuring the results, it would be helpful to provide a summary of the major theme following by sub-headings which describe each of the sub-themes. The sub-themes are currently mixed in within the major themes, which is not clear to the reader.
  3. The conceptual framework should be reported and discussed within the results, not the discussion.
  4. Implications and recommendations should come before the conclusion.

Author Response

Dear Reviewer 1,

Reviewer #1 Comments

I thank the author for taking the time to revise their manuscript. They have addressed most of the comments from the first round of revision. However, some minor amendments are still required.

AUTHOR’S REPLY: Thank you very much for the positive feedback and for some minor amendments that you suggested for the improvement of my work.

  1. In the introduction, the author now provides more context on the COVID-19 pandemic. However, now that the pandemic has ended, the introduction should reflect why it is important to ensure psychological well-being and emotional resilience in healthcare workers for post-pandemic recovery (e.g., lines 53-55).

AUTHOR’S REPLY: This has been addressed as suggested in Lines 57-59.

  1. When structuring the results, it would be helpful to provide a summary of the major theme following by sub-headings which describe each of the sub-themes. The sub-themes are currently mixed in within the major themes, which is not clear to the reader.

AUTHOR’S REPLY: This part of the Results section has been re-structured to clearly present the two major / primary themes with corresponding sub-themes in Lines 289-302.

  1. The conceptual framework should be reported and discussed within the results, not the discussion.

AUTHOR’S REPLY: The conceptual framework has been reported and discussed in the last part of the Results section, as suggested by the honorable Reviewer #1. In addition, the framework has been re-structured in a vertical alignment (top down), as suggested by the honorable Reviewer #2. Please see Lines 429-450.

4.Implications and recommendations should come before the conclusion.

AUTHOR’S REPLY: This has been implemented, as suggested in Lines 614-681.

Reviewer 2 Report (Previous Reviewer 2)

Comments and Suggestions for Authors

Thank you for the opportunity to provide my comments on your esteemed research. I have some minor comments based on my understanding of your research.
1. Title and Abstract
The authors can consider mentioning expatriate status as a key influencing factor to steer the readers.

2. Introduction
The manuscript can be strengthened by linking the experiences of nurses to the Job Demands-Resources (JD-R) model.
The "organisational stressors" can be demands, and the lack of family presence can be a diminished Resource.
Research questions can be better rephrased to capture the essence of "lived experience". For example, RQ2 can be rephrased like "How do frontline expatriate nurses navigate psychological challenges and foster resilience while caring for COVID-19 patients?". This shift in language aligns more closely with phenomenological inquiry.
3. Materials and Methods
The section is presented well with the required clarity and level of detail.

4. Results
Results are presented well.
Can the authors consider reorienting Figure 1, probably in a vertical alignment (top down), to enhance readability?
5. Discussion
Discussion regarding the increase in jobs during the pandemic can be helpful.
This can be linked with the JD-R model. Further, recent literature could be used to link the results of the present study to suggest handling similar situations.
Long-term psychological impact can also be mentioned. 

Author Response

Dear Reviewer 2,

Reviewer #2 Comments

Thank you for the opportunity to provide my comments on your esteemed research. I have some minor comments based on my understanding of your research.

AUTHOR’S REPLY: Thank you very much for the positive feedback and for some minor comments that you suggested to enhance my work.

  1. Title and Abstract
    The authors can consider mentioning expatriate status as a key influencing factor to steer the readers.

AUTHOR’S REPLY: This has been considered and added in Lines 28-29.

  1. Introduction
    The manuscript can be strengthened by linking the experiences of nurses to the Job Demands-Resources (JD-R) model. The "organisational stressors" can be demands, and the lack of family presence can be a diminished Resource.

AUTHOR’S REPLY: If the honorable Reviewer #2 may allow me not to implement this suggestion, I will highly appreciate it because this means making major revisions in this section and in the citation number order and referencing. Correspondingly, adding the Job Demands-Resources (JD-R) model in the Introduction section will require further revisions in the Discussion section. If the honorable Reviewer #2 will insist on implementing this, kindly know that I am very willing to comply.

Research questions can be better rephrased to capture the essence of "lived experience". For example, RQ2 can be rephrased like "How do frontline expatriate nurses navigate psychological challenges and foster resilience while caring for COVID-19 patients?". This shift in language aligns more closely with phenomenological inquiry.

AUTHOR’S REPLY: Thank you very much for this valuable suggestion. This has been implemented in Lines 118-121.

  1. Materials and Methods
    The section is presented well with the required clarity and level of detail.

AUTHOR’S REPLY: Thank you very much for the positive feedback.

  1. Results
    Results are presented well. Can the authors consider reorienting Figure 1, probably in a vertical alignment (top down), to enhance readability?

AUTHOR’S REPLY: This has been implemented. Also, Figure 1 has been transferred to the end part of the Results section as suggested by the honorable Reviewer #1. Please see in Lines 429-450.

  1. Discussion
    Discussion regarding the increase in jobs during the pandemic can be helpful. This can be linked with the JD-R model. Further, recent literature could be used to link the results of the present study to suggest handling similar situations. Long-term psychological impact can also be mentioned. 

AUTHOR’S REPLY: As I humbly requested to the honorable Reviewer #2 for me not to implement Job Demands-Resources (JD-R) model in the second round of revision, I have not implemented anything in this comment. Meanwhile, long-term psychological impact has been mentioned in Line/s 601. Thank you very much and Godspeed!

Reviewer 3 Report (Previous Reviewer 3)

Comments and Suggestions for Authors

Dear authors

Thank you for your careful and thorough revision of the manuscript. The new version shows substantial improvement in terms of methodological rigour, clarity, and alignment with qualitative reporting standards.

The abstract now includes a justification of the phenomenological approach and mention of rigour, which was previously missing.

The introduction has been strengthened with updated references and more articulated research questions. The rationale for focusing on expatriate nurses in Saudi Arabia is more convincing, although the justification for Husserlian phenomenology could still be framed more concisely.

The methods section has been expanded: the design is now explicitly identified as Husserlian descriptive phenomenology; reflexivity, rigour strategies, coder validation, and member checking are clearly described.

The results are supported by a demographic table, improving transferability. Quotations illustrate the themes effectively, though a few could be shortened to avoid repetition.

The discussion benefits from the addition of a conceptual framework and stronger links to existing literature, though integration with resilience/migration theory could still be deepened.

The limitations are more transparent, and a COREQ statement has been added.

Overall, the manuscript is much clearer, methodologically sound, and now close to publication standard. Only minor refinements remain (stylistic and interpretive), but the key reviewer concerns have been adequately addressed.

Best regards 

Reviewer

Author Response

Dear Reviewer 3,

Reviewer #3 Comments

Thank you for your careful and thorough revision of the manuscript. The new version shows substantial improvement in terms of methodological rigour, clarity, and alignment with qualitative reporting standards.

AUTHOR’S REPLY: Thank you very much for the positive feedback and for the minor suggestions to improve my work.

The abstract now includes a justification of the phenomenological approach and mention of rigour, which was previously missing.

AUTHOR’S REPLY: Thank you very much for the positive feedback.

The introduction has been strengthened with updated references and more articulated research questions. The rationale for focusing on expatriate nurses in Saudi Arabia is more convincing, although the justification for Husserlian phenomenology could still be framed more concisely.

AUTHOR’S REPLY: This has been addressed in Lines 110-112.

The methods section has been expanded: the design is now explicitly identified as Husserlian descriptive phenomenology; reflexivity, rigour strategies, coder validation, and member checking are clearly described.

AUTHOR’S REPLY: Thank you very much for the positive feedback.

The results are supported by a demographic table, improving transferability. Quotations illustrate the themes effectively, though a few could be shortened to avoid repetition.

AUTHOR’S REPLY: Some quotations have been shortened throughout the Results section.

The discussion benefits from the addition of a conceptual framework and stronger links to existing literature, though integration with resilience/migration theory could still be deepened.

AUTHOR’S REPLY: I respectfully value this suggestion of the honorable Reviewer #3 and humbly request not to implement it due to the utilization of Husserlian phenomenology approach.

The limitations are more transparent, and a COREQ statement has been added.

AUTHOR’S REPLY: Thank you very much for the positive feedback.

Overall, the manuscript is much clearer, methodologically sound, and now close to publication standard. Only minor refinements remain (stylistic and interpretive), but the key reviewer concerns have been adequately addressed.

AUTHOR’S REPLY: Thank you very much for the positive feedback.

This manuscript is a resubmission of an earlier submission. The following is a list of the peer review reports and author responses from that submission.

Round 1

Reviewer 1 Report

Comments and Suggestions for Authors

Thank you for the opportunity to review this manuscript. The author conducted semi-structured interviews with 17 frontline expatriate nurses in the Kingdom of Saudi Arabia, finding two key themes relating to the psychological effects and resilience. However, it is not clear what the present study adds to the abundance of literature relating to the psychological effects of COVID-19 on healthcare workers from around the globe. The two themes lack depth and require a more comprehensive analyses to develop nuanced themes which address the aims. The manuscript needs to be updated to reflect the current situation and to outline the importance of the findings in the current context, given that WHO declared the end of the public health emergency in May 2023. I have provided more detailed suggestions below:

Title/abstract

  • State in the title that the study is of frontline expatriate nurses in the Kingdom of Saudi Arabia
  • The background section is a bit repetitive, particularly lines 12 to 15. Consider condensing

Introduction

  • Much of the first and second paragraphs could be cut or reduced as the health impacts of COVID-19 are now widely known and this paper is focused on the experiences of frontline nurses
  • There is an abundance of literature on the psychological impacts of the pandemic on frontline nurses from countries beyond the KSA and this literature should be incorporated into the introduction
  • There is insufficient rationale for the focus on non-Saudi or expatriate nurses
  • More justification is needed as to why this research is novel and of timely importance, given that there is a lot of research in this area

Materials and methods

  • Check the wording of the following sentence, “those who met the eligibility criteria were approached for written consent from each of the participant”
  • It may be helpful to include the topic guide as an appendices

Results

  • Given that the two aims focused on understanding the psychological effects and strategies for coping, I feel that the theme names could be developed further to summarise what was found rather than just representing the two main aims
  • At the beginning of the results section, the author states that there were sub-themes, yet the sub-themes are not clear within each of the themes. For example, the author discusses findings relating to anxiety and depression, with quotes, yet the distinction between anxiety and depression is not clear. The sub-themes could be more nuanced, such as “psychological effects due to fear of contracting the virus” and/or “organisational stressors”
  • In the final paragraph, the author states, “the social stigma associated with the ongoing pandemic represents a significant factor influencing the mental wellbeing of nurses, given that it may contribute to an increased risk of virus transmission”. This paper needs to be updated to reflect the current situation, whereby the pandemic is no longer ongoing.

Discussion

  • As with the results section, the implications and recommendations need updating to reflect the current situation, given that WHO declared the end of the public health emergency in May 2023

Author Response

Dear Reviewer 1,

REVIEWER #1 COMMENTS

Comment: Thank you for the opportunity to review this manuscript. The author conducted semi-structured interviews with 17 frontline expatriate nurses in the Kingdom of Saudi Arabia, finding two key themes relating to the psychological effects and resilience. However, it is not clear what the present study adds to the abundance of literature relating to the psychological effects of COVID-19 on healthcare workers from around the globe. The two themes lack depth and require a more comprehensive analyses to develop nuanced themes which address the aims. The manuscript needs to be updated to reflect the current situation and to outline the importance of the findings in the current context, given that WHO declared the end of the public health emergency in May 2023. I have provided more detailed suggestions below:

AUTHOR’S REPLY: We highly appreciate the honorable Reviewer #1 for the comprehensive and constructive comments that helped improved my work. In the revised version of my work, there are significant contributions as addition to the abundance of literature to the psychological effects of COVID-19 on healthcare workers that are specifically unique to expatriate nurses in the country and in preparation for future pandemics, particularly in the last paragraph of the Introduction section (see Lines 124-146). May the revised version of my work merit your recommendation to the esteemed editor for its eventual publication in this prestigious MDPI journal, Healthcare. Thank you very much and Godspeed!

Title/abstract

  • State in the title that the study is of frontline expatriate nurses in the Kingdom of Saudi Arabia

AUTHOR’S REPLY: This has been added and stated in the title.

  • The background section is a bit repetitive, particularly lines 12 to 15. Consider condensing

AUTHOR’S REPLY: This has been revised in Lines 12-17.

Introduction

  • Much of the first and second paragraphs could be cut or reduced as the health impacts of COVID-19 are now widely known and this paper is focused on the experiences of frontline nurses

AUTHOR’S REPLY: This section has been reduced, as suggested, in Lines 43-60.

  • There is an abundance of literature on the psychological impacts of the pandemic on frontline nurses from countries beyond the KSA and this literature should be incorporated into the introduction

AUTHOR’S REPLY: This part has been strategically followed and revised, after searching for related studies having consistent findings with the current study across the world including China, Philippines, Portugal, Singapore, South Korea, and the United States of America in Lines 116-117.

  • There is insufficient rationale for the focus on non-Saudi or expatriate nurses

AUTHOR’S REPLY: This has been improved and put more emphasis on expatriate nurses in Lines 117-126.

  • More justification is needed as to why this research is novel and of timely importance, given that there is a lot of research in this area

AUTHOR’S REPLY: Additional justifications have been added in Lines 127-142.

Materials and methods

  • Check the wording of the following sentence, “those who met the eligibility criteria were approached for written consent from each of the participant”

AUTHOR’S REPLY: This has been revised, as suggested, in Lines 205-206. Thank you for noticing this.

  • It may be helpful to include the topic guide as an appendices

AUTHOR’S REPLY: May the honorable Reviewer #1 allow me to add the topic guide in the main text, specifically in Lines 223-260.

Results

  • Given that the two aims focused on understanding the psychological effects and strategies for coping, I feel that the theme names could be developed further to summarise what was found rather than just representing the two main aims

AUTHOR’S REPLY: Thank you and I have taken substantial considerations in revising the Results section based on this comment.

  • At the beginning of the results section, the author states that there were sub-themes, yet the sub-themes are not clear within each of the themes. For example, the author discusses findings relating to anxiety and depression, with quotes, yet the distinction between anxiety and depression is not clear. The sub-themes could be more nuanced, such as “psychological effects due to fear of contracting the virus” and/or “organisational stressors”

AUTHOR’S REPLY: Thank you for your valuable suggestion. I revised this part by using your suggested sub-theme as organizational stressors such as having anxiety, depression, fear and mental stress, and retained stigmatization as second sub-theme (see Lines 323-332).

  • In the final paragraph, the author states, “the social stigma associated with the ongoing pandemic represents a significant factor influencing the mental wellbeing of nurses, given that it may contribute to an increased risk of virus transmission”. This paper needs to be updated to reflect the current situation, whereby the pandemic is no longer ongoing.

AUTHOR’S REPLY: This has been revised, making it during the peak of the pandemic (see Lines 410-411).

Discussion

  • As with the results section, the implications and recommendations need updating to reflect the current situation, given that WHO declared the end of the public health emergency in May 2023

AUTHOR’S REPLY: This has been updated, as suggested, throughout the Discussion section (see Lines 463-625).

Reviewer 2 Report

Comments and Suggestions for Authors

Overall Comments

This manuscript presents a timely and important qualitative exploration of the psychological well-being and resilience of frontline expatriate nurses in the Kingdom of Saudi Arabia (KSA) during the COVID-19 pandemic. The study sheds light on the significant psychological toll experienced by this specific group of healthcare professionals and the coping mechanisms they employed. The paper is generally well-structured and clearly written. The topic is of considerable interest, particularly given the global reliance on expatriate healthcare workers. The findings contribute valuable insights into the lived experiences of these nurses. The suggestions below are intended to further strengthen the manuscript for publication.

Specific Comments by Section

Abstract The abstract provides a concise and accurate summary of the study's background, objectives, methods, results, and conclusions.

Suggestion: Consider very briefly mentioning the specific number of participants in the methods part of the abstract to give an immediate sense of the study's scale.

  1. Introduction The introduction effectively sets the stage by outlining the context of the COVID-19 pandemic, its impact on healthcare workers, and specifically, the gap in research concerning expatriate nurses in the KSA. The rationale for the study is well-articulated.

Suggestions for Improvement:

  • Research Questions: The current research questions are clear. To perhaps add a bit more nuance, the authors could consider framing a question that more directly explores the unique challenges faced by expatriate nurses compared to local nurses, if the data supports this. For example: "How did the expatriate status of frontline nurses shape their psychological experiences and access to resilience-building resources during the COVID-19 pandemic in KSA?" This might already be implicitly covered but making it explicit could be beneficial.
  • Literature: While the introduction cites relevant literature, incorporating a few more recent studies (post-2021/2022) on nurse well-being during later phases of the pandemic or comparative studies from other regions with high expatriate healthcare populations could enrich the background.
    • Example Literature Suggestion: Look for studies on "moral injury" in nurses during COVID-19, or studies comparing psychological impacts on local versus expatriate healthcare workers in similar contexts.
  • Theoretical Grounding: The introduction could be strengthened by briefly alluding to a theoretical framework that might underpin the concepts of psychological well-being and resilience in high-stress environments. For example, Lazarus and Folkman's Transactional Model of Stress and Coping could be relevant. A brief mention here could set the stage for a more detailed discussion later.
  1. Materials and Methods The methodology is appropriate for the research questions. The qualitative, phenomenological design is well-suited to exploring lived experiences. The sampling technique, data collection process, and ethical considerations are clearly described.

Suggestions for Improvement:

  • Detailing of Methodology:
    • Data Saturation: The authors mention reaching data saturation after the 17th participant. It would be useful to briefly describe how saturation was determined in practice (e.g., were new codes ceasing to emerge, were subsequent interviews yielding redundant information on key themes?).
    • Researcher Reflexivity: In qualitative research, the researcher's role and potential influence are important. A brief statement on researcher reflexivity – how the researcher's background or assumptions might have influenced the data collection and interpretation, and steps taken to mitigate bias – would strengthen the methodological rigor.
  • Rigor: The steps taken to ensure trustworthiness (credibility, dependability, confirmability, transferability) are well-described. The use of pilot interviews is a strength.
  1. Results The results are presented clearly, organized around two primary themes and their respective sub-themes. The use of direct quotations from participants is effective in illustrating the findings and giving voice to the nurses' experiences.

Suggestions for Improvement:

  • Nuance in Themes: The themes of "psychological effects" and "resilience" are appropriate. Within these, consider if the data allows for any further nuance. For instance, under "psychological effects," were there differences in the intensity or nature of anxiety/fear based on factors like years of experience, specific work unit, or personal family situation (e.g., family in KSA vs. family abroad)? This is not to suggest a quantitative analysis, but rather a qualitative exploration of such variations if present in the narratives.
  • Visual Representation: While not essential for qualitative work, the authors mention NVivo and word clouds for validation. If appropriate and insightful, a conceptual diagram illustrating the relationship between the main themes and sub-themes could be considered as a figure. This could help readers quickly grasp the core findings and their interconnections.
  1. Discussion The discussion effectively links the study's findings to existing literature, highlighting both consistencies and unique aspects. The exploration of each sub-theme with supporting literature is thorough.

Suggestions for Improvement:

  • Relating to Existing Theories: This section could be significantly enhanced by more explicitly connecting the findings to relevant psychological or sociological theories.
    • Example Theory Connection: The discussion of stress and coping strategies aligns well with Lazarus and Folkman's Transactional Model of Stress and Coping. The authors could discuss how nurses appraised the stressors (e.g., threat of infection, workload) and then engaged in problem-focused or emotion-focused coping (e.g., seeking accurate information, family support, maintaining health).
    • The theme of stigmatization could be linked to Goffman's theory of stigma, discussing how labels and social reactions impacted the nurses.
    • The concept of resilience could be discussed in the context of theories like the Salutogenic Model (Antonovsky), focusing on factors that support health and well-being rather than just the presence of stressors.
  • Emerging Trends/Gaps: Consider discussing the findings in light of the evolving nature of the pandemic. Were there shifts in psychological impact or coping strategies as the pandemic progressed (e.g., pre-vaccine vs. post-vaccine availability, different waves of infection)?
  • Framework Suggestion: The authors could propose or adapt a simple conceptual framework based on their findings. This framework could visually represent the interplay between pandemic-related stressors, the psychological effects on expatriate nurses, the moderating role of their expatriate status, their coping strategies/resilience factors, and the resulting well-being outcomes. This would be a valuable contribution.
  1. Conclusions The conclusions are well-supported by the results and summarize the key takeaways of the study effectively.

Suggestion: Briefly reiterate the specific vulnerability of expatriate nurses in the concluding remarks, emphasizing why targeted interventions for this group are particularly crucial.

  1. Implications and Recommendations This section provides practical and actionable recommendations for nursing practice, administration, education, and future research. The suggestions are thoughtful and well-grounded in the study's findings.

Suggestions for Improvement:

  • Specificity of Interventions: When suggesting targeted interventions, perhaps add a bit more specificity. For example, for "organizational support," what kind of support was most desired or deemed most effective by the nurses (e.g., more flexible scheduling, mental health days, better communication channels)?
  • Policy Implications: Consider explicitly mentioning any broader policy implications, such as the need for national or institutional policies that specifically address the well-being and support of expatriate healthcare workers during public health emergencies.

Author Response

Dear Reviewer 2,

Overall Comments:

This manuscript presents a timely and important qualitative exploration of the psychological well-being and resilience of frontline expatriate nurses in the Kingdom of Saudi Arabia (KSA) during the COVID-19 pandemic. The study sheds light on the significant psychological toll experienced by this specific group of healthcare professionals and the coping mechanisms they employed. The paper is generally well-structured and clearly written. The topic is of considerable interest, particularly given the global reliance on expatriate healthcare workers. The findings contribute valuable insights into the lived experiences of these nurses. The suggestions below are intended to further strengthen the manuscript for publication.

AUTHOR’S REPLY: Thank you very much for your valuable and constructive comments. I extensively revised my work based on them. I hope that the revisions I made are acceptable which will help convince the editor for the eventual acceptance of the revised version of my work. If not, kindly know that I am very willing to comply further with any additional feedback for improvement. More power and Godspeed!

Specific Comments by Section

Abstract The abstract provides a concise and accurate summary of the study's background, objectives, methods, results, and conclusions.

AUTHOR’S REPLY: Thank you very much for this positive feedback.

Suggestion: Consider very briefly mentioning the specific number of participants in the methods part of the abstract to give an immediate sense of the study's scale.

  1. Introduction The introduction effectively sets the stage by outlining the context of the COVID-19 pandemic, its impact on healthcare workers, and specifically, the gap in research concerning expatriate nurses in the KSA. The rationale for the study is well-articulated.

AUTHOR’S REPLY: Thank you very much for this positive feedback.

Suggestions for Improvement:

  • Research Questions: The current research questions are clear. To perhaps add a bit more nuance, the authors could consider framing a question that more directly explores the unique challenges faced by expatriate nurses compared to local nurses, if the data supports this. For example: "How did the expatriate status of frontline nurses shape their psychological experiences and access to resilience-building resources during the COVID-19 pandemic in KSA?" This might already be implicitly covered but making it explicit could be beneficial.

AUTHOR’S REPLY: This has been added, as suggested. Please see Lines 155-157.

  • Literature: While the introduction cites relevant literature, incorporating a few more recent studies (post-2021/2022) on nurse well-being during later phases of the pandemic or comparative studies from other regions with high expatriate healthcare populations could enrich the background.
    • Example Literature Suggestion: Look for studies on "moral injury" in nurses during COVID-19, or studies comparing psychological impacts on local versus expatriate healthcare workers in similar contexts.

AUTHOR’S REPLY: Several studies have been added to this effect in Lines 108-142.

  • Theoretical Grounding: The introduction could be strengthened by briefly alluding to a theoretical framework that might underpin the concepts of psychological well-being and resilience in high-stress environments. For example, Lazarus and Folkman's Transactional Model of Stress and Coping could be relevant. A brief mention here could set the stage for a more detailed discussion later.

AUTHOR’S REPLY: Thank you for this comment, but after following the suggestion of the honorable Reviewer #2, I identified the phenomenological design that guided this study by employing Husserlian’s phenomenological approach. Based on this approach, the inclusion and utilization of any theoretical framework were not included in alignment with Husserl's stance that no a priori phenomenological or theoretical framework should direct the phenomenological investigation (Lopez & Willis, 2004; Shorey & Ng, 2022).

Lopez, K. A., & Willis, D. G. (2004). Descriptive versus interpretive phenomenology: their contributions to nursing knowledge. Qualitative Health Research14(5), 726–735. https://doi.org/10.1177/1049732304263638

Shorey, S., & Ng, E. D. (2022). Examining characteristics of descriptive phenomenological nursing studies: A scoping review. Journal of Advanced Nursing78(7), 1968–1979. https://doi.org/10.1111/jan.15244

  1. Materials and Methods The methodology is appropriate for the research questions. The qualitative, phenomenological design is well-suited to exploring lived experiences. The sampling technique, data collection process, and ethical considerations are clearly described.

AUTHOR’S REPLY: Thank you very much for this positive feedback.

Suggestions for Improvement:

  • Detailing of Methodology:
    • Data Saturation: The authors mention reaching data saturation after the 17th participant. It would be useful to briefly describe how saturation was determined in practice (e.g., were new codes ceasing to emerge, were subsequent interviews yielding redundant information on key themes?).

AUTHOR’S REPLY: This is done and added, as suggested, in Lines 185-187.

    • Researcher Reflexivity: In qualitative research, the researcher's role and potential influence are important. A brief statement on researcher reflexivity – how the researcher's background or assumptions might have influenced the data collection and interpretation, and steps taken to mitigate bias – would strengthen the methodological rigor.

AUTHOR’S REPLY: I have added this, as suggested, in Lines 263-271.

  • Rigor: The steps taken to ensure trustworthiness (credibility, dependability, confirmability, transferability) are well-described. The use of pilot interviews is a strength.

AUTHOR’S REPLY: Thank you very much for this positive feedback.

  1. Results The results are presented clearly, organized around two primary themes and their respective sub-themes. The use of direct quotations from participants is effective in illustrating the findings and giving voice to the nurses' experiences.

AUTHOR’S REPLY: Thank you very much for this positive feedback.

Suggestions for Improvement:

  • Nuance in Themes: The themes of "psychological effects" and "resilience" are appropriate. Within these, consider if the data allows for any further nuance. For instance, under "psychological effects," were there differences in the intensity or nature of anxiety/fear based on factors like years of experience, specific work unit, or personal family situation (e.g., family in KSA vs. family abroad)? This is not to suggest a quantitative analysis, but rather a qualitative exploration of such variations if present in the narratives.

AUTHOR’S REPLY: Based on this comment, I sincerely apologize that only personal family situation has been addressed in Lines 346-351.

  • Visual Representation: While not essential for qualitative work, the authors mention NVivo and word clouds for validation. If appropriate and insightful, a conceptual diagram illustrating the relationship between the main themes and sub-themes could be considered as a figure. This could help readers quickly grasp the core findings and their interconnections.

AUTHOR’S REPLY: May I humbly request that the honorable Reviewer #2 would not include this, as it is not essential for qualitative work. If insisted, I am very willing to comply further. Thank you very much for your kindest consideration.

  1. Discussion The discussion effectively links the study's findings to existing literature, highlighting both consistencies and unique aspects. The exploration of each sub-theme with supporting literature is thorough.

AUTHOR’S REPLY: Thank you very much for this positive feedback.

Suggestions for Improvement:

  • Relating to Existing Theories: This section could be significantly enhanced by more explicitly connecting the findings to relevant psychological or sociological theories.
    • Example Theory Connection: The discussion of stress and coping strategies aligns well with Lazarus and Folkman's Transactional Model of Stress and Coping. The authors could discuss how nurses appraised the stressors (e.g., threat of infection, workload) and then engaged in problem-focused or emotion-focused coping (e.g., seeking accurate information, family support, maintaining health).

AUTHOR’S REPLY: As justified above, by adhering to the key features of Husserlian’s phenomenological approach, the inclusion and utilization of any theoretical framework were not included in alignment with Husserl's stance that no a priori phenomenological or theoretical framework should direct the phenomenological investigation (Lopez & Willis, 2004; Shorey & Ng, 2022).

    • The theme of stigmatization could be linked to Goffman's theory of stigma, discussing how labels and social reactions impacted the nurses.

AUTHOR’S REPLY: Also for this, as justified above, by adhering to the key features of Husserlian’s phenomenological approach, the inclusion and utilization of any theoretical framework were not included in alignment with Husserl's stance that no a priori phenomenological or theoretical framework should direct the phenomenological investigation (Lopez & Willis, 2004; Shorey & Ng, 2022).

    • The concept of resilience could be discussed in the context of theories like the Salutogenic Model (Antonovsky), focusing on factors that support health and well-being rather than just the presence of stressors.

AUTHOR’S REPLY: Also for this, as justified above, by adhering to the key features of Husserlian’s phenomenological approach, the inclusion and utilization of any theoretical framework were not included in alignment with Husserl's stance that no a priori phenomenological or theoretical framework should direct the phenomenological investigation (Lopez & Willis, 2004; Shorey & Ng, 2022).

  • Emerging Trends/Gaps: Consider discussing the findings in light of the evolving nature of the pandemic. Were there shifts in psychological impact or coping strategies as the pandemic progressed (e.g., pre-vaccine vs. post-vaccine availability, different waves of infection)?

AUTHOR’S REPLY: This has been addressed and added, as suggested, in Lines 621-625.

  • Framework Suggestion: The authors could propose or adapt a simple conceptual framework based on their findings. This framework could visually represent the interplay between pandemic-related stressors, the psychological effects on expatriate nurses, the moderating role of their expatriate status, their coping strategies/resilience factors, and the resulting well-being outcomes. This would be a valuable contribution.

AUTHOR’S REPLY: This has been addressed and added, as suggested, in Lines 626-647.

  1. Conclusions The conclusions are well-supported by the results and summarize the key takeaways of the study effectively.

AUTHOR’S REPLY: Thank you very much for this positive feedback.

Suggestion: Briefly reiterate the specific vulnerability of expatriate nurses in the concluding remarks, emphasizing why targeted interventions for this group are particularly crucial.

AUTHOR’S REPLY: This has been added and emphasized in Lines 659-663.

  1. Implications and Recommendations This section provides practical and actionable recommendations for nursing practice, administration, education, and future research. The suggestions are thoughtful and well-grounded in the study's findings.

AUTHOR’S REPLY: Thank you very much for this positive feedback.

Suggestions for Improvement:

  • Specificity of Interventions: When suggesting targeted interventions, perhaps add a bit more specificity. For example, for "organizational support," what kind of support was most desired or deemed most effective by the nurses (e.g., more flexible scheduling, mental health days, better communication channels)?

AUTHOR’S REPLY: Thank you very much for this valuable suggestion. This has been added in Lines 675-683.

  • Policy Implications: Consider explicitly mentioning any broader policy implications, such as the need for national or institutional policies that specifically address the well-being and support of expatriate healthcare workers during public health emergencies.

AUTHOR’S REPLY: Thank you very much for this valuable suggestion. This has been added in Lines 725-731.

Reviewer 3 Report

Comments and Suggestions for Authors

Dear authors,

Thank you for the opportunity to review this manuscript. The study addresses an important and timely topic, exploring the psychological impact of COVID-19 on expatriate frontline nurses in Saudi Arabia. The qualitative approach offers valuable insights, and the focus on resilience adds depth o the current literature.

I commend the authors for their work and for addressing a crucial gap concerning the well-being of expatriate nurses during the pandemic.

Below, I provide detailed comments and suggestions to strengthen the manuscript further.

COMMENT 1. Abstract: Justification of phenomenological approach and rigor criteria missing

Lines: 8–32. The abstract outlines the design and main findings, but does not justify the use of a phenomenological approach

COMMENT 2. Introduction: Literature gap and conceptual framework are insufficiently defined. Lines: 12–17, 100–102. While the lack of studies on expatriate nurses is mentioned, it is not clear what is missing in previous literature and why phenomenology is needed.

What exactly have previous studies overlooked? Why is a phenomenological lens essential here? What theoretical frameworks (e.g., resilience, stress, migration) inform your approach?

COMMENT 3. Introduction: Research questions need to be more specific and explicitly qualitative. Lines: 16–17, 106–110

The study aim is relatively generic; the qualitative research question is not clearly stated. Reword the questions: “How do frontline expatriate nurses experience and make sense of the psychological impact of caring for COVID-19 patients in KSA?”

COMMENT 4. Methods – Research Design: Lack of detail on phenomenological model and researcher reflexivity

Lines: 113–117 Comment: The phenomenological approach is mentioned but not specified (e.g., Husserlian, Heideggerian, descriptive, interpretive), and there’s no discussion of researcher positioning or reflexivity. Lines: 170–194 . Guba and Lincoln’s four criteria are mentioned, but there is little concrete description of how each was addressed—especially transferability (thick description?), credibility (triangulation, member checking?), and confirmability.

Provide specific examples for each criterion (e.g., “Transferability was supported by detailed contextual descriptions and a demographic table; credibility by peer debriefing…”).

COMMENT 9. Methods – Data Analysis: Coder validation and member checking

Lines: 195–203. Comment:Thematic analysis and NVivo are described, but it’s unclear whether coding was double-checked, how discrepancies were resolved, or if member checking occurred.

State: “A subset of transcripts was coded independently by two researchers. Discrepancies were discussed until consensus was achieved.”

COMMENT 10. Results: Demographic profile of participants is lacking

Lines: 204–221. There is little to no demographic or contextual description of the sample, which is necessary for transferability.

Add a paragraph or table outlining participants’ age range, years of experience, country of origin, unit of work, etc.

COMMENT 12. Discussion: Need to relate findings more to theory and qualitative literatura Lines: 337–496

The discussion is comprehensive in citing literature, but it is not sufficiently linked to conceptual frameworks (resilience, trauma, migration) or the unique value of a qualitative/phenomenological approach.

COMMENT 13. Discussion: Qualitative limitations are not explicitly stated

Lines: 497–507

There is little discussion of the limitations inherent in qualitative research (e.g., context-specific findings, potential for researcher bias).

Add: “As in all qualitative research, findings are context-bound and may not be generalizable. The researcher’s positionality and reflexivity are acknowledged as influencing data collection and interpretation.”

COMMENT 15. Reporting Standards: Declare use of COREQ

Placement: Before “Author Contributions” section

There is no statement about reporting standards (e.g., COREQ checklist) followed.

Add: “This study adhered to the COREQ (Consolidated Criteria for Reporting Qualitative Research) checklist for qualitative studies (Tong et al., 2007).”

Best regards

Author Response

Dear Reviewer 3,

Comment: Thank you for the opportunity to review this manuscript. The study addresses an important and timely topic, exploring the psychological impact of COVID-19 on expatriate frontline nurses in Saudi Arabia. The qualitative approach offers valuable insights, and the focus on resilience adds depth o the current literature.

I commend the authors for their work and for addressing a crucial gap concerning the well-being of expatriate nurses during the pandemic.

AUTHOR’S REPLY: Thank you very much for highlighting this valuable contribution to my work concerning the wellbeing of expatriate nurses during the COVID-19 pandemic and more importantly to your positive constructive feedback. Your detailed comments helped me improve the revised version of my work, of which I earnestly hope would also help convince the esteemed editor to decide the acceptance of my work for publication. More power and Godspeed!

Below, I provide detailed comments and suggestions to strengthen the manuscript further.

COMMENT Abstract: Justification of phenomenological approach and rigor criteria missing

AUTHOR’S REPLY: This has been added, as suggested, in Lines 15, 20-22.

Lines: 8–32. The abstract outlines the design and main findings, but does not justify the use of a phenomenological approach

AUTHOR’S REPLY: This has been justified in Line/s 15.

COMMENT Introduction: Literature gap and conceptual framework are insufficiently defined. Lines: 12–17, 100–102. While the lack of studies on expatriate nurses is mentioned, it is not clear what is missing in previous literature and why phenomenology is needed.

What exactly have previous studies overlooked? Why is a phenomenological lens essential here? What theoretical frameworks (e.g., resilience, stress, migration) inform your approach?

AUTHOR’S REPLY: Thank you for this comment, but after following the suggestion of the honorable Reviewer #2, I identified the phenomenological design that guided this study by employing Husserlian’s phenomenological approach. Based on this approach, the inclusion and utilization of any theoretical framework were not included in alignment with Husserl's stance that no a priori phenomenological or theoretical framework should direct the phenomenological investigation (Lopez & Willis, 2004; Shorey & Ng, 2022).

Lopez, K. A., & Willis, D. G. (2004). Descriptive versus interpretive phenomenology: their contributions to nursing knowledge. Qualitative Health Research14(5), 726–735. https://doi.org/10.1177/1049732304263638

Shorey, S., & Ng, E. D. (2022). Examining characteristics of descriptive phenomenological nursing studies: A scoping review. Journal of Advanced Nursing78(7), 1968–1979. https://doi.org/10.1111/jan.15244

COMMENT Introduction: Research questions need to be more specific and explicitly qualitative. Lines: 16–17, 106–110

The study aim is relatively generic; the qualitative research question is not clearly stated. Reword the questions: “How do frontline expatriate nurses experience and make sense of the psychological impact of caring for COVID-19 patients in KSA?”

AUTHOR’S REPLY: This has been revised, as suggested, in Lines 150-152. Thank you very much for this valuable suggestion.

COMMENT Methods – Research Design: Lack of detail on phenomenological model and researcher reflexivity

AUTHOR’S REPLY: Adequate details have been added for phenomenological model in Lines 160-168, and researcher reflexivity in Lines 263-271.

COMMENT Lines: 113–117 Comment: The phenomenological approach is mentioned but not specified (e.g., Husserlian, Heideggerian, descriptive, interpretive), and there’s no discussion of researcher positioning or reflexivity.

AUTHOR’S REPLY: It has been specified as Husserlian in Line/s 161.

COMMENT Lines: 170–194 . Guba and Lincoln’s four criteria are mentioned, but there is little concrete description of how each was addressed—especially transferability (thick description?), credibility (triangulation, member checking?), and confirmability.

Provide specific examples for each criterion (e.g., “Transferability was supported by detailed contextual descriptions and a demographic table; credibility by peer debriefing…”).

AUTHOR’S REPLY: These parts have been revised, as suggested, in Lines 272-296.

COMMENT Methods – Data Analysis: Coder validation and member checking

Lines: 195–203. Comment:Thematic analysis and NVivo are described, but it’s unclear whether coding was double-checked, how discrepancies were resolved, or if member checking occurred.

State: “A subset of transcripts was coded independently by two researchers. Discrepancies were discussed until consensus was achieved.”

AUTHOR’S REPLY: This has been added, as suggested, in Lines 305-307. Thank you very much for this valuable suggestion.

COMMENT Results: Demographic profile of participants is lacking

Lines: 204–221. There is little to no demographic or contextual description of the sample, which is necessary for transferability.

Add a paragraph or table outlining participants’ age range, years of experience, country of origin, unit of work, etc.

AUTHOR’S REPLY: This has been added in Lines 309-312. Correspondingly, Table 1 has been added as Demographic Profile of Participants in the part of Page 7.

COMMENT Discussion: Need to relate findings more to theory and qualitative literatura Lines: 337–496

The discussion is comprehensive in citing literature, but it is not sufficiently linked to conceptual frameworks (resilience, trauma, migration) or the unique value of a qualitative/phenomenological approach.

AUTHOR’S REPLY: This is not attainable in the current study upon employing Husserlian’s phenomenological approach as justified above. Meanwhile, the researcher added a simple conceptual framework, as suggested by the honorable Reviewer #2, and considered as valuable contribution of the study findings, in Lines 626-647.

COMMENT Discussion: Qualitative limitations are not explicitly stated

Lines: 497–507

There is little discussion of the limitations inherent in qualitative research (e.g., context-specific findings, potential for researcher bias).

Add: “As in all qualitative research, findings are context-bound and may not be generalizable. The researcher’s positionality and reflexivity are acknowledged as influencing data collection and interpretation.”

AUTHOR’S REPLY: This has been added, as suggested, in Lines 631-633. Thank you very much for this valuable suggestion.

COMMENT Reporting Standards: Declare use of COREQ

Placement: Before “Author Contributions” section

There is no statement about reporting standards (e.g., COREQ checklist) followed.

Add: “This study adhered to the COREQ (Consolidated Criteria for Reporting Qualitative Research) checklist for qualitative studies (Tong et al., 2007).”

AUTHOR’S REPLY: This has been added, as suggested, in Lines 732-733. Also, the citation has been added to the Reference List.